# Validation of the REGARDS Severe Sepsis Risk Score

**DOI:** 10.3390/jcm7120536

**Published:** 2018-12-11

**Authors:** Henry E. Wang, John P. Donnelly, Sachin Yende, Emily B. Levitan, Nathan I. Shapiro, Yuling Dai, Hong Zhao, Gerardo Heiss, Michelle Odden, Anne Newman, Monika Safford

**Affiliations:** 1Department of Emergency Medicine, The University of Texas Health Science Center at Houston, Houston, TX 77030, USA; 2Department of Learning Health Sciences, University of Michigan, Ann Arbor, MI 48109, USA; jpdonn@med.umich.edu; 3Department of Critical Care Medicine, University of Pittsburgh, Veterans Affairs Pittsburgh Healthcare System, Pittsburgh, PA 15213, USA; yendes@upmc.edu; 4Department of Epidemiology, University of Alabama at Birmingham, Birmingham, AL 35294, USA; eblevitan@uab.edu (E.B.L.); ydai@uabmc.edu (Y.D.); hozhao@uab.edu (H.Z.); 5Department of Emergency Medicine, Beth Israel Deaconess Medical Center, Boston, MA 02215, USA; nshapiro@bidmc.harvard.edu; 6Department of Epidemiology, University of North Carolina at Chapel Hill, Chapel Hill, NC 27599, USA; gerardo_heiss@unc.edu; 7Department of Health Research and Policy, Stanford University, Stanford, CA 94305, USA; modden@stanford.edu; 8Department of Epidemiology, University of Pittsburgh, Pittsburgh, PA 15213, USA; newmana@edc.pitt.edu; 9Department of Medicine, Weill-Cornell Medical Center, New York, NY 10065, USA; mms9024@med.cornell.edu

**Keywords:** sepsis, infections, risk prediction, outcomes

## Abstract

There are no validated systems for characterizing long-term risk of severe sepsis in community-dwelling adults. We tested the ability of the REasons for Geographic and Racial Differences in Stroke-Severe Sepsis Risk Score (REGARDS-SSRS) to predict 10-year severe sepsis risk in separate cohorts of community-dwelling adults. We internally tested the REGARDS-SSRS on the REGARDS-Medicare subcohort. We then externally validated the REGARDS-SSRS using (1) the Cardiovascular Health Study (CHS) and (2) the Atherosclerosis Risk in Communities (ARIC) cohorts. Participants included community-dwelling adults: REGARDS-Medicare, age ≥65 years, *n* = 9522; CHS, age ≥65 years, *n* = 5888; ARIC, age 45–64 years, *n* = 11,584. The primary exposure was 10-year severe sepsis risk, predicted by the REGARDS-SSRS from participant sociodemographics, health behaviors, chronic medical conditions and select biomarkers. The primary outcome was first severe sepsis hospitalizations, defined as the concurrent presence of ICD-9 discharge diagnoses for a serious infection and organ dysfunction. Median SSRS in the cohorts were: REGARDS-Medicare 11 points (IQR 7–16), CHS 10 (IQR 6–15), ARIC 7 (IQR 5–10). Severe sepsis incidence rates were: REGARDS-Medicare 30.7 per 1000 person-years (95% CI: 29.2–32.2); CHS 11.9 (10.9–12.9); ARIC 6.8 (6.3–7.3). SSRS discrimination for first severe sepsis events were: REGARDS-Medicare C-statistic 0.704 (95% CI: 0.691–0.718), CHS 0.696 (0.675–0.716), ARIC 0.697 (0.677–0.716). The REGARDS-SRSS may potentially play a role in identifying community-dwelling adults at high severe sepsis risk.

## 1. Introduction

The sepsis syndrome is a major public health problem responsible for over 850,000 emergency department visits, 200,000 deaths and $16.7 billion in medical expenditures in the United States (US) each year [1,2,3,4]. Current scientific and clinical initiatives focus almost entirely on the early recognition and treatment of acute sepsis [5,6]. However, the onset of acute disease is often influenced by underlying precursors. For example, acute myocardial infarction and stroke were once viewed as acute conditions triggered by random exposures or processes that could not be anticipated or prevented. The recognition of myocardial infarction and stroke as the end result of underlying chronic medical conditions led to new strategies of cardiovascular risk prediction, management and prevention, contributing to dramatic declines in heart disease and stroke mortality [7,8,9]. Like myocardial infarction and stroke in the past, relatively little attention has been paid to the factors that may heighten an individual’s risk of acute sepsis. An understanding of the factors that influence an individual’s baseline sepsis risk could provide new opportunities for mitigating the societal burden of sepsis [10].

Fundamental steps in disease prevention include identifying vulnerable individuals and quantifying their degree of risk. In prior studies, we used the population-based REasons for Geographic and Racial Differences in Stroke (REGARDS) cohort to show that sepsis hospitalizations are predictable and associated with sociodemographic factors, health behaviors, chronic diseases, and perturbations in the immune system [11]. We used these findings to derive a risk prediction rule—the REGARDS Severe Sepsis Risk Score (REGARDS-SSRS)—quantifying the 10-year risk of severe sepsis hospitalizations [12] (Table 1 and Appendix Table A1).

To validate their accuracy, robustness and generalizability, risk prediction rules must be tested in independent cohorts [13]. In this study we first tested the ability of the REGARDS-SSRS to identify severe sepsis events in a subcohort of REGARDS participants enrolled in Medicare. We then sought to externally validate the REGARDS-SSRS in two large population-based cohorts: The Cardiovascular Health Study (CHS) and the Atherosclerotic Risk in Communities (ARIC) Study.

## 2. Experimental Section

### 2.1. Study Design and Overview

We used a population-based cohort design. This study was approved by the Institutional Review Boards of the University of Alabama at Birmingham and the University of Texas Health Science Center at Houston.

While we originally derived the REGARDS-SSRS using severe sepsis events identified by adjudicated review of medical records, there were no other existing cohorts with similarly identified severe sepsis events available for validation of the prediction rule. However, many population-based cohorts have discharge diagnoses available for all hospitalizations, providing an alternate strategy for identifying severe sepsis events. We therefore developed strategies to validate the REGARDS-SSRS in discharge-diagnosis based data sets. 

As an important initial step, we first internally applied the SSRS to the REGARDS-Medicare cohort; this subcohort linked Medicare-enrolled REGARDS participants with their respective Medicare claims. This initial analysis allowed us to evaluate the ability of the SRSS to predict sepsis cases in the REGARDS cohort using hospital discharge diagnoses. We then sought to externally validate the SSRS using the CHS and ARIC cohorts, two of the most widely recognized longitudinal population-based cohorts in the US.

### 2.2. Study Setting and Participants—The REGARDS-Medicare, CHS and ARIC Cohorts

REGARDS is one of the nation’s largest ongoing population-based cohorts [14]. Consisting of 30,239 community-dwelling adults aged ≥45 years from across the United States, the REGARDS cohort was designed to evaluate the predictors of racial and geographic differences in stroke mortality. Recruitment of REGARDS participants occurred between January 2003 and October 2007 with ascertainment of baseline participant information including medical history, functional status, health behaviors, physical characteristics, physiologic measures, current medications, diet, family history of diseases, psychosocial factors and prior residences. The study also obtained comprehensive biologic specimens at the time of enrollment. Among REGARDS participants, 42% are African American, 45% are male, and 69% are over 60 years old.

The REGARDS-Medicare subcohort consists of 9,522 REGARDS participants who were enrolled in Medicare during 2003–2012 [15,16,17,18]. For this study, we limited the analysis to REGARDS participants age ≥65 years enrolled in Medicare Parts A (inpatient services) and B (outpatient services) throughout the entire REGARDS follow-up period 2003–2012. We excluded those with Medicare Part C coverage, who transitioned into or out of Medicare coverage during the study period. The parent REGARDS study identified hospitalizations through adjudicated chart review of medical records for each participant-reported health event. In contrast, the REGARDS-Medicare subcohort identified hospitalization events using discharge diagnoses for each claim. The REGARDS-Medicare linkage strategy has been previously described [18,19,20]. The characteristics of the REGARDS-Medicare subcohort are similar to participants in the Medicare national sample [18]. While the REGARDS-Medicare subcohort contains individuals used in the original SSRS derivation, the method of severe sepsis event identification is different, entailing use of discharge diagnoses. REGARDS included individuals ≥45 years old while Medicare was limited to persons ≥65 years old; we accepted this limitation as there was no large data source characterizing insurance claims for all REGARDS participants.

The Cardiovascular Health Study (CHS) is a population-based longitudinal study of coronary heart disease and stroke in adults aged ≥65 years [21]. Since 1987 the cohort CHS recruited a total of 5888 participants from Forsyth County (North Carolina), Sacramento County (California), Washington County (Maryland) and Pittsburgh (Pennsylvania). The study conducted extensive physical and laboratory evaluations at baseline. Follow-up of participants entailed semi-annual clinic examination (enrollment through 1999) and phone contact (enrollment through present). Identification of hospitalization events in CHS occurred through semi-annual interviews, additional self-reports by participants, and periodic surveillance of Medicare records. The study obtained discharge summaries and diagnoses for all hospitalizations.

The Atherosclerosis Risk in Communities Study (ARIC) is a prospective epidemiologic study conducted in Forsyth County (North Carolina), Jackson (Mississippi), Minneapolis (Minnesota), and Washington County (Maryland) [22]. In 1987 ARIC recruited a cohort a total of 15,792 individuals aged 45–64 years, conducting an extensive examination, and determination of medical, social, and demographic data. Study personnel abstracted hospitalization records (including diagnosis and procedure codes) for events identified through annual telephone contact with study participants or surveillance in study hospitals.

We chose CHS and ARIC cohorts because of their longitudinal design, completeness of follow-up, the availability of comprehensive discharge diagnoses for all hospitalizations, and the depth of available baseline variables and biomarkers needed to calculate the SSRS for each participant.

### 2.3. Exposure—Calculation of REGARDS-Severe Sepsis Risk Score

The REGARDS-Severe Sepsis Risk Score (SSRS) characterizes long-term severe sepsis risk using combinations of individual sociodemographics, health behaviors, chronic medical conditions, and biomarkers. (Table 1) Derivation and internal validation of the SSRS has been previously described [12]. Briefly, for each participant reported serious infection hospitalizations, two trained abstractors independently reviewed the corresponding medical records for the first 28-h of hospitalization to confirm the presence of a serious infection, sepsis and organ dysfunction. Using Cox regression, we identified clinical characteristics associated with severe sepsis events, converting the regression model into a 0–45 point severe sepsis risk score (SSRS).

While our original derivation effort included a prediction model for sepsis events (Table A1), for this study we focused on the model identifying severe sepsis events because of its higher discrimination and clearer link with the contemporary definition of sepsis as infection plus organ dysfunction [23]. We also focused on the version of the SSRS incorporating biomarkers (hsCRP, Cystatin-C, eGFR, ACR) because of its higher discrimination.

We calculated REGARDS-SSRS scores for each participant in the REGARDS-Medicare, CHS and ARIC cohorts using baseline comorbidity data available for each participant. For CHS and ARIC, we used previously published serum creatinine adjustments in order to calibrate these values for use with the CKD-EPI equation in the calculation of estimated glomerular filtration rates (eGFR) [24]. Baseline serum creatinine values for REGARDS participants were previously IDMS standardized [25].

For the ARIC cohort, ACR and Cystatin-C were not available until participant visit (examination) No. 4 (1996–1998). Therefore, for ARIC, we used data collected at visit 4 to calculate risk score points. The visit 4 examination date was used as the baseline for all ARIC participants in calculations of time to sepsis, with individuals excluded if the visit 4 examination was not completed for any reason or if sepsis occurred before this date.

### 2.4. Outcomes—Identification of Severe Sepsis Events

The primary outcome was first severe sepsis hospitalization. Following the method of Angus, et al., we defined severe sepsis as hospitalizations with the concurrent presence of discharge diagnoses for a serious infection and organ dysfunction [2] (Table A2 and Table A3). We identified severe sepsis events during the following follow-up periods: REGARDS-Medicare 2003–2012, CHS 1989–2002 and ARIC 1996–2008. Although ARIC enrolled subjects starting in 1987, we limited the analysis to sepsis events identified during 1996–2008 because certain exposure variables (ACR, Cystatin-C and history of atrial fibrillation) were not available until participant examination No. 4 (1996–1998).

### 2.5. Data Analysis

We compared participant characteristics between the REGARDS-Medicare, CHS and ARIC cohorts. We assessed the distribution of the SSRS in each cohort using quantile plots and histograms.

To test the ability of the SSRS to detect discharge-diagnosis-identified severe sepsis events, we first internally applied the SSRS to the REGARDS-Medicare subcohort. We used the original SSRS model and point system—we did not derive a new risk score for the analysis. Using Cox proportional hazards models, we determined severe sepsis incidence and relative hazards across risk groups (very low, low, medium, high, and very high, as previously reported in the initial SSRS derivation effort). To assess calibration, we conducted a validation in the manner described by Royston [26]. We estimated predicted and observed survival functions for SSRS risk groups from Cox models, with the smoothed baseline function estimated using fractional polynomials. We assessed goodness of fit using the Groennesby and Borgan score test (Stata ‘stcoxgof’ command) and reported the observed and predicted number of events over predicted risk deciles [27]. We determined discrimination using Harrell’s C-statistic [28]. Lastly, we reported the percent positive, the true positive rate, and the true negative rate over the full ten years of follow-up.

We then externally validated application of the SSRS to the ARIC and CHS data sets. We followed a similar strategy as with the REGARDS-Medicare subcohort, examining incidence and relative hazards across risk groups, observed and predicted survival functions and events, goodness-of-fit and discrimination. For the ARIC cohort we excluded participants who died or experienced severe sepsis events prior to 1996.

In the primary analysis, we explicitly coded missing exposure variables as “normal”. We conducted all analyses using Stata 13.1 (Stata, Inc., College Station, TX, USA).

## 3. Results

From the REGARDS-Medicare, CHS and ARIC cohorts, we included 9522, 5888, and 11,584 participants, respectively (Table 2). While the REGARDS-Medicare and CHS cohorts included only persons ≥65 years of age, 60.4% of the ARIC cohort participants were <65 years old at baseline (Table 2). The REGARDS-Medicare cohort contained a slightly higher proportion of males and blacks. Current tobacco use was more common in ARIC participants. The REGARDS-Medicare cohort exhibited a higher comorbid burden than the CHS and ARIC cohorts. History of atrial fibrillation, coronary artery disease, diabetes, deep vein thrombosis, hypertension and stroke were most prevalent in the REGARDS-Medicare cohort. Chronic lung disease was most prevalent in the CHS cohort. Peripheral artery disease was most prevalent in ARIC. Prevalence of obesity was similar between the cohorts. Among biomarkers, CHS participants exhibited a slightly higher prevalence of abnormal hsCRP and eGFR. Elevated Cystatin-C and ACR were most common among REGARDS-Medicare participants.

Median SSRS were: REGARDS-Medicare 11 points (IQR 7–16), CHS 10 (IQR 6–15), ARIC 7 (IQR 5–10). Quantile plot and histograms patterns suggested similar SSRS distributions were between the REGARDS-Medicare and CHS cohorts (Figure 1). The ARIC cohort generally exhibited lower SSRS risk scores. The number of events and incidence rates of severe sepsis events in each cohort were: REGARDS-Medicare, *n* = 1593, IR 30.7 per 1000 person-years (95% CI: 29.2–32.2); CHS *n* = 586, IR 11.9 (10.9–12.9); ARIC *n* = 735, IR 6.8 (6.3–7.3) (Table 3).

The SSRS showed adequate discrimination in the REGARDS-Medicare subcohort; C-statistic 0.704 (95% CI: 0.691–0.718) (Table 3). The SSRS showed good calibration in the REGARDS-Medicare subcohort across risk groups based on plots of observed and predicted survival functions (Figure 2). The SSRS also demonstrated good fit in the REGARDS-Medicare subcohort.

In the ARIC and CHS cohorts, the hazard ratios for severe sepsis events were similar across SSRS risk categories (Table 3). The SSRS showed adequate discrimination; CHS 0.696 (0.675–0.716), ARIC 0.697 (0.677–0.716) (Table 4). In ARIC and CHS, the SSRS showed good calibration across risk groups based on plots of observed and predicted survival functions (Figure 2). For the CHS cohort, the p-value for goodness of fit was statistically significant, but the observed and predicted number of events across deciles of predicted risk were similar for all cohorts. The application of the SSRS to ARIC demonstrated good fit. The observed and predicted number of events across deciles of predicted risk were similar for the ARIC and CHS cohorts (Table 4).

Across all cohorts, less than 6% of participants in the very low risk group developed severe sepsis over ten years of follow-up, while over 20% of participants in the very high risk group had an event (Table 5). In addition, less than 7% of all participants with severe sepsis in REGARDS-Medicare and CHS were classified as very low risk. In ARIC, a higher proportion of severe sepsis cases were classified as low risk, but this may not be comparable to the other cohorts due to differences in baseline risk. Less than 15% of participants without severe sepsis were classified as very high risk across all cohorts.

## 4. Discussion

We previously derived and internally validated the REGARDS-SRSS using severe sepsis events identified by adjudicated review of medical records [12]. In this study, we verified the ability of the REGARDS-SSRS to predict long-term severe sepsis events identified through Medicare claims in the REGARDS-Medicare subcohort. We also externally validated the SSRS using the CHS and ARIC cohorts. Our results illustrate the robustness and generalizability of the REGARDS-SSRS, supporting its potential application in the prediction of individual and community severe sepsis risk.

The potential application of the REGARDS-SSRS is in frameworks that may be unfamiliar to acute care practitioners. Existing decision rules predict sepsis or adverse outcomes among acutely hospitalized patients [29]. In contrast, the SSRS predicts long-term severe sepsis risk among community-dwelling adults. Our study underscores the concept that risk prediction and prevention could play pivotal roles in reducing the public health burden of sepsis [10]. For example, sepsis entails infection complicated by systemic inflammation and organ dysfunction, and individuals with chronic conditions may be less able to tolerate or respond to this stress. The SSRS related severe sepsis risk to the presence of chronic conditions such as chronic lung disease, peripheral artery disease, tobacco use, coronary artery disease, obesity, hypertension, deep vein thrombosis and chronic kidney disease; optimal management of these conditions might mitigate long-term sepsis risk. Statin therapy has been proposed for reduce cardiovascular risk among individuals with elevated hsCRP [30]. Similar approaches may be possible with sepsis.

Individuals with known high SSRS risk may merit personalized approaches to prevention, acute sepsis care and even health education; for example, these individuals may receive increased emphasis on vaccines and tailored education on recognizing early signs of sepsis. The SSRS may also have important public health applications, identifying community groups or clusters for organized deployment of sepsis prevention measures such as those described above. Additional study must determine if modification of these risk factors in fact alters sepsis risk.

An important aspect of our study was the application of the REGARDS-SSRS to the detection of sepsis using hospital discharge diagnoses. We originally derived the SSRS using severe sepsis events identified through structured medical record review and adjudication. While best practices would have entailed external validation with an analogously structured data set, there are presently no cohorts with severe sepsis events identified in a similar manner. Instead we validated a pragmatic and efficient alternate approach using severe sepsis events identified through Medicare claims and discharge diagnoses. This study highlights the robust nature of the REGARDS-SSRS. National health care data sets in Denmark and Taiwan have been used to characterize sepsis epidemiology in these respective countries [31,32]. Our findings suggest that potential additional inquiry or validation of the SSRS could occur using these other claims- or discharge-diagnosis based data sets.

Experts customarily recommend that clinical decision rules exhibit a discrimination (C-statistic) value of at least 0.70 on derivation and validation [28,33]. The C-statistic values observed in this study (REGARDS-Medicare 0.704, CHS 0.696, ARIC 0.697) fell just short of this threshold. One potential explanation may entail the inherent limitations of the Angus severe sepsis criteria. Discharge diagnoses do not incorporate physiologic or laboratory measures and may be affected by biases in the documented discharge diagnoses. The Angus taxonomy presumes a clinical connection between a serious infection diagnosis and an organ dysfunction diagnosis. While the original SSRS validation identified only community-acquired severe sepsis events, discharge diagnoses cannot differentiate between community-acquired, hospital-acquired or healthcare-associated severe sepsis. In a separate effort using a hospital discharge data set with “present-on-admission” flags for each diagnosis, we found that Angus severe sepsis hospitalizations encompassed approximately 62.8% community-acquired, 25.9% hospital-acquired and 11.3% healthcare associated severe sepsis cases [34]. Additional opportunities for improving the discrimination of the SSRS may encompass the identification of additional relevant clinical variables or biomarkers.

This study has limitations. Our original efforts resulted in separate risk prediction models for both sepsis (infection + ≥2 systemic inflammatory response syndrome criteria) and severe sepsis (sepsis + ≥1 SOFA organ failure point) [12]. In the current study we focused on severe sepsis as the primary endpoint because it best aligned with the discharge data available in the test cohorts. Furthermore, current Sepsis-3 consensus guidelines advocate defining sepsis as the combination of [infection + organ dysfunction], a definition that closely aligns with the definition for severe sepsis [23]. We previously noted that the predictors of the REGARDS-SRS and SSRS were nearly identical, and that 70% of the derivation sepsis events also fulfilled severe sepsis criteria. Some cases presenting to the hospital with sepsis may have later fulfilled severe sepsis criteria. In the original derivation effort, we did not focus on septic shock due to the smaller number of events in the REGARDS cohort.

In the original REGARDS-SSRS derivation we developed models both with and without biomarker variables. For the current study, we focused on the REGARDS-SSRS with biomarker elements due its higher discriminatory power and the availability of identical biomarkers in the test cohorts.

The original REGARDS-sepsis effort identified sepsis hospitalizations through manual chart review and included only events occurring within the first 28-h of hospitalization. The current analysis utilized an alternate approach entailing ICD-9 discharge diagnoses appearing in Medicare claims, and may have encompassed sepsis events at any point of hospitalization. Comparison of the agreement between manual chart review and discharge diagnoses is the objective of a separate analysis and not the focus of the current study. Prior studies suggest the low sensitivity and high sensitivity of discharge diagnoses for identifying severe sepsis [35]. Also, changes in coding practices and reimbursement incentives may influence sepsis coding [36]. These points are evident by the higher severe sepsis incidence seen in current study. However, even with these limitations, we were able to apply and validate the SSRS using these alternate sepsis identification methods. We acknowledge that the ARIC and CHS data used for validation are 10–20 years old; validation with more recent data is an important future direction.

The current effort using the REGARDS-Medicare, CHS and ARIC cohorts illustrates potential alternate strategies for detecting sepsis events. For example, Medicare claims data may capture the most significant health events for an individual. Additional study with cohorts linked to hospital discharge data may lead to additional insights regarding sepsis event detection and classification. The original SSRS was based upon the outcome of community-acquired severe sepsis. In the current study we could not differentiate community-acquired from hospital-acquired severe sepsis because the study data sets do not indicate diagnoses that were present on admission. Additional validation with a cohort incorporating present-on-admission diagnosis flags is an important future goal. While some of the biomarkers in the SSRS are relatively new (specifically, Cystatin-C), the assay can be potentially incorporated into clinical practice.

The REGARDS cohort included only African Americans and whites ≥45 years. REGARDS also did not include nursing home patients, who may be more vulnerable to sepsis than community dwelling individuals. While we used biomarkers that were readily available in REGARDS. Other biomarkers may have potentially improved model discrimination. We could not account for changes in participant characteristics over time. We did not have information on the presence of select immunosuppressive comorbidities such as human immunodeficiency virus infection or liver disease. The risk score does not include markers of future infection risk such as prior infections, antimicrobial use, or use of urinary catheters. Re-derivation of the SSRS with a broader cohort is necessary to evaluate the impact of these factors. While we conceptualized the SSRS as having roles in primary prevention, further study is also needed to confirm if risk modification in fact leads to reduced sepsis risk. We did not account for the competing risk of death because this would have complicated the interpretation of sepsis risk.

## 5. Conclusions

In this study we tested the ability of the REGARDS-Severe Sepsis Risk Score to predict 10-year risk of severe sepsis when applied to the REGARDS-Medicare, CHS and ARIC cohorts. The REGARDS-SSRS may potentially play a role in community sepsis prevention or mitigation efforts.

## Figures and Tables

**Figure 1 jcm-07-00536-f001:**
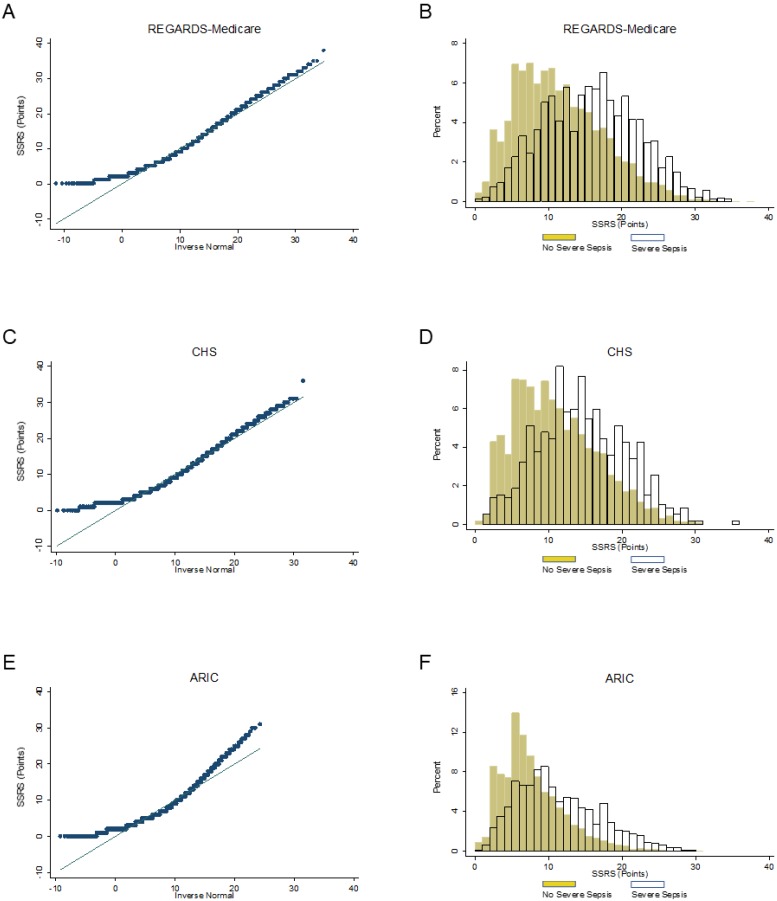
Normality (panels **A**, **C**, **E**) and distribution (panels **B**, **D**, **F**) of the REGARDS Severe Sepsis Risk Score in REGARDS-Medicare, CHS and ARIC cohorts. REGARDS = Reasons for Geographic and Racial Differences in Stroke. CHS = Cardiovascular Health Study. ARIC = Atherosclerosis Risk in Communities.

**Figure 2 jcm-07-00536-f002:**
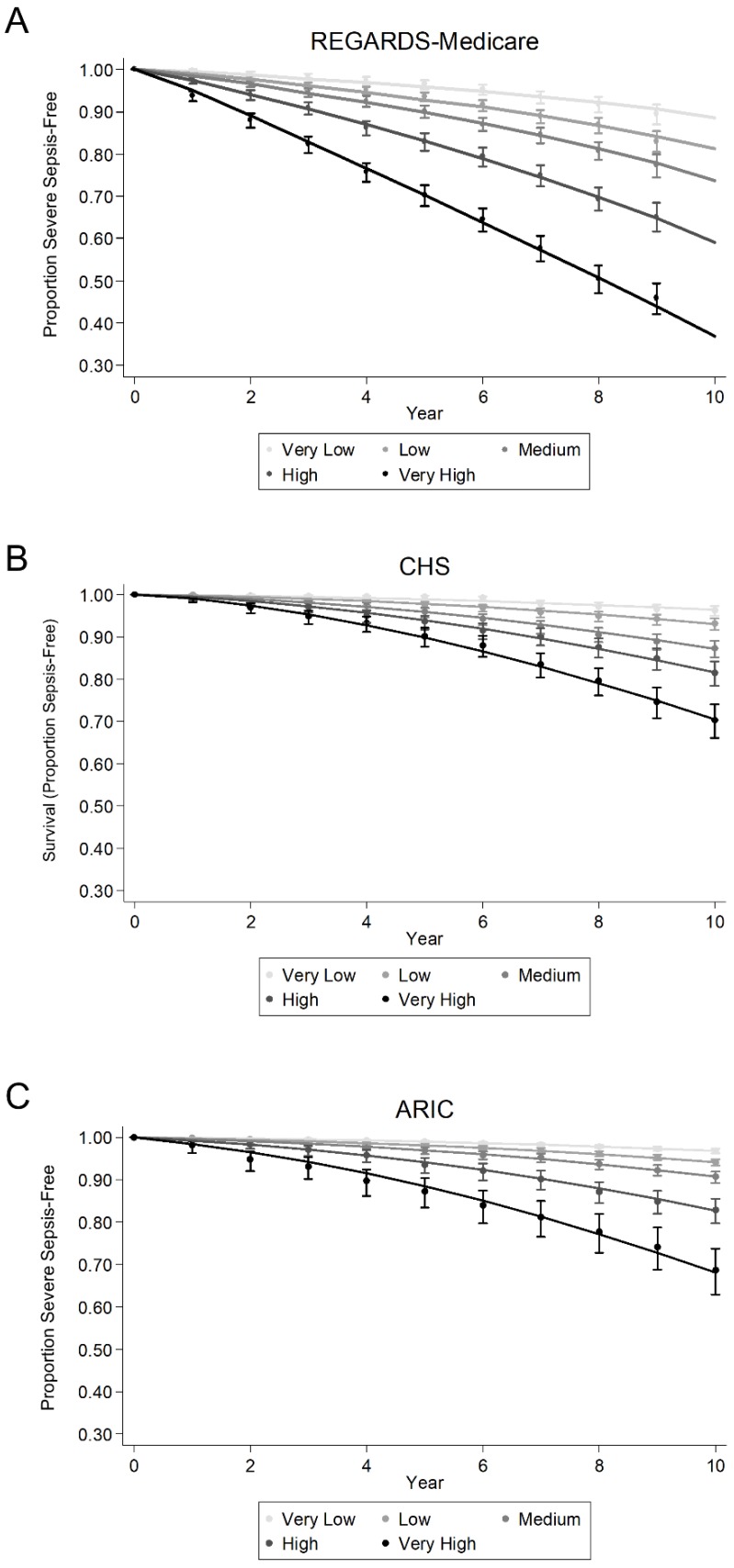
Calibration of the REGARDS Severe Sepsis Risk Score in the REGARDS-Medicare (**A**), CHS (**B**) and ARIC (**C**) cohorts. REGARDS = Reasons for Geographic and Racial Differences in Stroke. CHS = Cardiovascular Health Study. ARIC = Atherosclerosis Risk in Communities.

**Table 1 jcm-07-00536-t001:** The REGARDS Severe Sepsis Risk Score (SSRS).

Variable	SSRS Points
Chronic lung disease	5
Age ≥75 years	4
Peripheral Artery Disease	3
Diabetes	3
Male Sex	3
Tobacco use (current)	2
White Race	2
Stroke	2
Atrial Fibrillation	2
Coronary artery disease	2
Deep Vein Thrombosis	2
Obesity	1
Hypertension	1
Cystatin-C ≥1.11 mg/dL	5
hsCRP >3.0 mg/dL	3
ACR ≥30 mcg/mg	3
eGFR (creatinine) <60 mL/min/1.73 m^2^	2
TOTAL SSRS POINTS	45

hsCRP = high sensitivity C-reactive protein. ACR = Albumin-to-creatinine ratio, eGFR = estimated glomerular filtration rate.

**Table 2 jcm-07-00536-t002:** Baseline participant characteristics by cohort.

Characteristic	Internal Application	External Validation	
	REGARDS-Medicare (*n* = 9522)	CHS (*n* = 5888)	ARIC (*n* = 11,584)
**Age**			
<65 years	0	0	6998 (60.4)
65–74	6120 (64.3)	3894 (66.1)	4583 (39.6)
≥75	3402 (35.7)	1994 (33.9)	3 (0.0)
**Gender**			
Male	4692 (49.3)	2495 (42.4)	5106 (44.1)
Female	4830 (50.7)	3393 (57.6)	6478 (55.9)
**Race**			
White	6440 (67.6)	4925 (83.6)	8899 (76.8)
Black	3082 (32.4)	963 (16.4)	2685 (23.2)
**Tobacco Use**			
Current	910 (9.6)	700 (11.9)	1700 (14.7)
No Current Use	8612 (90.4)	5188 (88.1)	9884 (85.3)
**Chronic Medical Conditions**			
Atrial Fibrillation	1025 (10.8)	236 (4.0)	324 (2.8)
Chronic Lung Disease	1013 (10.6)	915 (15.5)	929 (8.0)
Coronary Artery Disease	2364 (24.8)	1,154 (19.6)	965 (8.3)
Diabetes	2276 (23.9)	989 (16.8)	1924 (16.6)
Deep Vein Thrombosis	648 (6.8)	312 (5.3)	348 (3.0)
Hypertension	6200 (65.1)	2619 (44.5)	5510 (47.6)
Obesity	4746 (49.8)	2747 (46.7)	7592 (65.5)
Peripheral Artery Disease	275 (2.9)	120 (2.0)	385 (3.3)
Stroke	796 (8.4)	249 (4.2)	265 (2.3)
**High-Sensitivity C-Reactive Protein**			
>3.0 mg/dL	3449 (36.2)	2481 (42.1)	4909 (42.4)
≤3.0 mg/dL	6073 (63.8)	3407 (57.9)	6675 (57.6)
**Cystatin-C**			
≥1.11 mg/dL	3486 (36.6)	1569 (26.7)	1504 (13.0)
<1.11 mg/dL	6036 (63.4)	4319 (73.4)	10,080 (87.0)
**eGFR (creatinine)**			
<60 mL/min/1.73 m^2^	1723 (18.1)	1368 (24.1)	1112 (9.6)
≥60 mL/min/1.73 m^2^	7799 (81.9)	4520 (76.8)	10,472 (90.4)
**Albumin-to-Creatinine Ratio**			
≥30 mcg/mg	1612 (16.9)	654 (11.1)	956 (8.3)
<30 mcg/mg	7910 (83.1)	5234 (88.9)	10,628 (91.8)
**SSRS Risk Group**			
Very Low (<6 Points)	1616 (17.0)	1137 (19.3)	3761 (32.5)
Low (6–9 Points)	2308 (24.2)	1582 (26.9)	4388 (37.9)
Medium (10–13 Points)	2127 (22.3)	1358 (23.1)	2037 (17.6)
High (14–17 Points)	1687 (17.7)	988 (16.8)	900 (7.8)
Very High (>17 Points)	1784 (18.7)	823 (14.0)	498 (4.3)

REGARDS = Reasons for Geographic and Racial Differences in Stroke. CHS = Cardiovascular Health Study. ARIC = Atherosclerosis Risk in Communities.

**Table 3 jcm-07-00536-t003:** Severe sepsis incidence for each cohort, overall and stratified by REGARDS-Severe Sepsis Risk Score risk category.

Risk Group	Internal Application		External Validation			
	REGARDS-Medicare (*n* = 9522)		CHS (*n* = 5888)		ARIC (*n* = 11,584)	
	Severe Sepsis Events (*n*) IR (95% CI)	HR (95% CI)	Severe Sepsis Events (*n*) IR (95% CI)	HR (95% CI)	Severe Sepsis Events (*n*) IR (95% CI)	HR (95% CI)
**Full Cohort**	1593 30.7 (29.2–32.2)	---	586 11.9 (10.9–12.9)	---	735 6.8 (6.3–7.3)	---
**Stratified by REGARDS-SSRS Risk Category**						
Very Low (<6 Points)	96 9.9 (8.1–12.1)	Reference	39 3.6 (2.6–4.9)	Reference	114 3.1 (2.6–3.8)	Reference
Low (6–9 Points)	230 17.0 (14.9–19.3)	1.73 (1.36–2.19)	99 6.9 (5.7–8.4)	1.94 (1.34–2.82)	208 5.0 (4.3–5.7)	1.60 (1.27–2.01)
Medium (10–13 Points)	299 24.8 (22.2–27.8)	2.53 (2.01–3.19)	143 12.8 (10.8–15.0)	3.71 (2.61–5.29)	179 9.7 (8.4–11.2)	3.15 (2.49–3.99)
High (14–17 Points)	374 42.0 (37.9–46.5)	4.36 (3.48–5.45)	138 18.4 (15.6–21.8)	5.51 (3.86–7.86)	118 15.6 (13.0–18.7)	5.20 (4.02–6.73)
Very High (>17 Points)	594 76.8 (70.9–83.3)	8.27 (6.66–10.26)	167 30.2 (25.9–35.1)	9.46 (6.67–13.41)	116 33.7 (28.1–40.4)	11.98 (9.24–15.52)

REGARDS = Reasons for Geographic and Racial Differences in Stroke. CHS = Cardiovascular Health Study. ARIC = Atherosclerosis Risk in Communities. IR = incidence rate per 1000 person-years. HR = hazard ratio. CI = confidence interval.

**Table 4 jcm-07-00536-t004:** Discrimination, calibration, and goodness-of-fit of the REGARDS-Severe Sepsis Risk Score for each cohort.

Measure	REGARDS-Medicare	CHS	ARIC
Discrimination—C-statistic (95% CI)	0.704 (0.691–0.718)	0.696 (0.675–0.716)	0.697 (0.677–0.716)
Decile of Predicted Risk (Observed/Predicted Events)			
1	60/59.9	29/30.7	41/41.0
2	89/95.1	29/47.1	31/23.2
3	97/108.1	30/25.5	42/47.9
4	80/68.1	50/55.4	56/61.9
5	150/147.5	27/28.4	54/56.6
6	149/162.9	80/61.1	44/48.3
7	179/172.3	78/69.5	54/48.2
8	195/183.3	67/61.2	93/90.6
9	302/298.8	80/83.9	124/116.8
10	292/297.0	116/123.2	196/200.4
Goodness of Fit—Grønnesby and Borgan test (Chi-square; *p*-value)	7.66 (0.57)	18.95 (0.03)	5.72 (0.77)

REGARDS = Reasons for Geographic and Racial Differences in Stroke. CHS = Cardiovascular Health Study. ARIC = Atherosclerosis Risk in Communities.

**Table 5 jcm-07-00536-t005:** Percentage positive with severe sepsis, true positive rate, and true negative rate by REGARDS-Severe Sepsis Risk Score risk category.

Risk Group	REGARDS-Medicare			CHS			ARIC		
	PP (% with Severe Sepsis)	TPR (% of Cases)	TNR (% of Non-Cases)	PP (% with Severe Sepsis)	TPR (% of Cases)	TNR (% of Non-Cases)	PP (% with Severe Sepsis)	TPR (% of Cases)	TNR (% of Non-Cases)
Very Low (<6 Points)	5.9	6.0	19.2	3.4	6.7	20.7	3.0	15.5	33.6
Low (6–9 Points)	10.0	14.4	26.2	6.3	16.9	28.0	4.7	28.3	38.5
Medium (10–13 Points)	14.1	18.8	23.1	10.5	24.4	22.9	8.8	24.4	17.1
High (14–17 Points)	22.2	23.5	16.6	14.0	23.5	16.0	13.1	16.1	7.2
Very High (>17 Points)	33.3	37.3	15.0	20.3	28.5	12.4	23.3	15.8	3.5

REGARDS = Reasons for Geographic and Racial Differences in Stroke. CHS = Cardiovascular Health Study. ARIC = Atherosclerosis Risk in Communities. PP = percent positive. TPR = true positive rate. TNR = true negative rate.

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
