# Peer review of "Validation of the REGARDS Severe Sepsis Risk Score"

_jcm, 2018, doi:10.3390/jcm7120536_

Reviewer 1 Report

Dear author

I read the manuscript with interest, and it is important to predict the long-term risk of severe sepsis (or sepsis) in community setting. The authors validated the REGARDS Severe Sepsis Risk using 3 different cohorts. However, there were some limitations to interpret the results as authors’ comments. The diagnosis of severe sepsis using the discharge diagnosis may be inaccurate though the identification of sepsis is a key element in this study. And the important risk factors for infection (e.g., immunocompromised state, the presence of malignancy, medication) were not considered for the analysis. These factors might be associated with the low discrimination power.

My detailed concern about this study is two.

1. In author's first paper (CCM, 2016), severe sepsis incidence rate was 6.2 per 1,000 person-years. But, this study shows the severe sepsis incidence rate was 30.7 per 1,000 person-years in same cohort (REGARDS-Medicare). As author's comments, many hospital-acquired severe sepsis events might be included for analysis. In terms of that, this validation study for the REGARDS Severe Sepsis Risk Score may be inaccurate even if the large cohorts was used for analysis.Including the discharge code, more clinical information such as the use of antibiotics, blood culture, WBC count at admission could be considered for the diagnosis of the community-setting severe sepsis

2. The key point of sepsis diagnosis is the presence of the infection. The severe sepsis is defined by the organ dysfunction and infection.When we make s scoring system, variables of scoring system could explain the scientific relation logically not statistically.

I don't know why these variables such as Atrial fibrillation, coronary artery disease, hypertension, ACR, and eGFR are risk factors of sepsis/severe sepsis. These variables may be associated with the organ dysfunction regardless of infection.I'm afraid clinicians stretch the meaning of the results.

Author Response

1. In author's first paper (CCM, 2016), severe sepsis incidence rate was 6.2 per 1,000 person-years. But, this study shows the severe sepsis incidence rate was 30.7 per 1,000 person-years in same cohort (REGARDS-Medicare). As author's comments, many hospital-acquired severe sepsis events might be included for analysis. In terms of that, this validation study for the REGARDS Severe Sepsis Risk Score may be inaccurate even if the large cohorts was used for analysis. Including the discharge code, more clinical information such as the use of antibiotics, blood culture, WBC count at admission could be considered for the diagnosis of the community-setting severe sepsis

RESPONSE: Thank you for this astute observation. Two factors likely influence sepsis incidence in this paper:

        a) The use of discharge diagnoses for identifying severe sepsis events (vs. manually adjudicated chart review) likely resulted in higher and more complete             event detection. The under-detection of events has always been an acknowledged limitation of adjudicated chart review. Unfortunately, measures             needed for defining clinical criteria were unavailable for the cohorts used for external validation. We overcame this limitation through the use of               administrative claims data.

  b) We externally applied the SSRS to two different cohorts with different baseline risk (CHS and ARIC). It is not surprising that disease incidence may vary between cohorts.

We have acknowledged these issues in the Limitations para 3; Prior studies suggest the low sensitivity and high sensitivity of discharge diagnoses for identifying severe sepsis.[35] Also, changes in coding practices and reimbursement incentives may influence sepsis coding.[36] These points are evident by the higher severe sepsis incidence seen in current study.

         2. The key point of sepsis diagnosis is the presence of the infection. The severe sepsis is defined by the organ dysfunction and infection. When we              make s scoring system, variables of scoring system could explain the scientific relation logically not statistically.

             I don't know why these variables such as Atrial fibrillation, coronary artery disease, hypertension, ACR, and eGFR are risk factors of sepsis/severe             sepsis. These variables may be associated with the organ dysfunction regardless of infection. I'm afraid clinicians stretch the meaning of the results.

RESPONSE: This is exactly our overarching points – that chronic conditions may in fact act as risk factors for severe sepsis. Sepsis involve dysregulated systemic response to infection. Individuals with vascular conditions may be less able to tolerate/respond to the systemic stress. We clarified this conceptual point in the Discussion paragraph 2. In addition, each of the variables included in the risk score were independently associated with sepsis as defined by infection with clinical criteria indicative of organ dysfunction.

Reviewer 2 Report

The manuscript, “Validation of the REGARDS Severe Sepsis Risk Score” is a follow up study from their previous manuscript entitled, “Derivation of a Novel Risk Prediction Scores for Community-Acquired Sepsis and Severe Sepsis”, which develops various models to predict long-term risks of future sepsis and severe sepsis events among community-dwelling adults. The previous study utilized the REasons for Geographic And Racial Differences in Stroke (REGARDS) cohort, which is one of the largest longitudinal population-based cohorts in the US, to identify sociodemographic factors, modifiable health behavior, preventable and treatable chronic medical conditions, and biomarkers independently associated with sepsis over a 10 year period. The authors used this cohort to identify risk factors involved with hospitalization for sepsis or severe sepsis. The current manuscript attempts to externally validate their model using the Medicare only patients in the REGARDS cohort along with two other population-based cohorts: the Cardiovascular Health Study (CHS) and the Atherosclerotic Risk in Communities (ARIC) Study. The authors chose these cohorts due to their longitudinal design, completeness of follow up, the availability of comprehensive discharge diagnoses, and the depth of available variables.
These studies in combination provide an interesting addition to the current understanding of risk factors for sepsis and severe sepsis. This was a necessary validation study, given that the original model was derived from a stroke risk factor cohort. However, here are a few questions/comments that still need to be addressed prior to publication.
1. The authors utilize discharge diagnosis to identify the patients with severe sepsis used in this validation study. As mentioned in the limitations section, this does not discriminate between patients who presented to the hospital with sepsis and who developed sepsis while in the hospital for other reasons. Since the hypothesis for this study is that the SSRS predicts hospitalization for sepsis in community dwellers, this is a significant limitation. Have the authors attempted to use admission diagnosis of severe sepsis to identify patients and subsequently validate their model? Given that the Cstat falls below 0.7 for both CHS and ARIC, this may improve the robustness of their model.
2.  Another major limitation is that the authors only used the prior definition of severe sepsis to identify patients to use in this validation study (sepsis, as defined by infection plus two or more SIRS criteria, plus evidence of organ dysfunction). Since the previous definition of severe sepsis is more similar to the current definition of sepsis, the authors chose to only include patients with “severe sepsis” or “no severe sepsis”. How do the authors identify and capture patients who meet the current criteria for septic shock? This could significantly skew the data if septic shock patients are missed or not included.
3. Another significant limitation is that the authors attempt to validate their SSRS model using cohorts that are 10-20 years old (CHS from 1987-2002 and ARIC from 1996-2008), which is before the original definitions of sepsis were even founded. Since the authors use discharge diagnoses to identify patients, can the authors describe how they justified the use of these cohorts in this context and how they identified sepsis diagnoses before the original consensus definition that was formulated in 1991?

Author Response

REVIEWER 2

Comments and Suggestions for Authors

The manuscript, “Validation of the REGARDS Severe Sepsis Risk Score” is a follow up study from their previous manuscript entitled, “Derivation of a Novel Risk Prediction Scores for Community-Acquired Sepsis and Severe Sepsis”, which develops various models to predict long-term risks of future sepsis and severe sepsis events among community-dwelling adults. The previous study utilized the REasons for Geographic And Racial Differences in Stroke (REGARDS) cohort, which is one of the largest longitudinal population-based cohorts in the US, to identify sociodemographic factors, modifiable health behavior, preventable and treatable chronic medical conditions, and biomarkers independently associated with sepsis over a 10 year period. The authors used this cohort to identify risk factors involved with hospitalization for sepsis or severe sepsis. The current manuscript attempts to externally validate their model using the Medicare only patients in the REGARDS cohort along with two other population-based cohorts: the Cardiovascular Health Study (CHS) and the Atherosclerotic Risk in Communities (ARIC) Study. The authors chose these cohorts due to their longitudinal design, completeness of follow up, the availability of comprehensive discharge diagnoses, and the depth of available variables.

These studies in combination provide an interesting addition to the current understanding of risk factors for sepsis and severe sepsis. This was a necessary validation study, given that the original model was derived from a stroke risk factor cohort. However, here are a few questions/comments that still need to be addressed prior to publication.

1.    The authors utilize discharge diagnosis to identify the patients with severe sepsis used in this validation study. As mentioned in the limitations section, this does not discriminate between patients who presented to the hospital with sepsis and who developed sepsis while in the hospital for other reasons. Since the hypothesis for this study is that the SSRS predicts hospitalization for sepsis in community dwellers, this is a significant limitation. Have the authors attempted to use admission diagnosis of severe sepsis to identify patients and subsequently validate their model? Given that the Cstat falls below 0.7 for both CHS and ARIC, this may improve the robustness of their model.

RESPONSE: We appreciate the suggestion. We naturally wished for admission diagnoses to identify severe sepsis events, but the study data sets (REGARDS-Medicare, ARIC, CHS) do not differentiate diagnoses that are present on admission from other diagnoses. To the best of our knowledge, hospital-based  discharge data sets are the only data sources organized in this format. Thus, we regret that cannot analyze the data as suggested. We amplified this point in the Limitations para 4.

2.    Another major limitation is that the authors only used the prior definition of severe sepsis to identify patients to use in this validation study (sepsis, as defined by infection plus two or more SIRS criteria, plus evidence of organ dysfunction). Since the previous definition of severe sepsis is more similar to the current definition of sepsis, the authors chose to only include patients with “severe sepsis” or “no severe sepsis”. How do the authors identify and capture patients who meet the current criteria for septic shock? This could significantly skew the data if septic shock patients are missed or not included.

RESPONSE: We appreciate the reviewer’s insights. As the reviewer has anticipated, our work in this area has evolved over the last 10 years, during which time the definitions of sepsis have evolved. We refrained from predicting septic shock in the original derivation because of the smaller number of septic shock events in the REGARDS cohort. While we agree with the reviewer’s sentiment, we feel that deriving a new prediction score for septic shock is beyond the scope of the project – this goal would require rederivation of a new septic shock prediction score from the original data set. Furthermore, septic shock is most accurately classified using clinical data such as systolic blood pressure rather than discharge diagnoses, as is the approach in the current validation effort. Therefore, we feel that introducing septic shock into the current analysis would be very confusing and distracting. We added comment to Limitations para 1.

3.    Another significant limitation is that the authors attempt to validate their SSRS model using cohorts that are 10-20 years old (CHS from 1987-2002 and ARIC from 1996-2008), which is before the original definitions of sepsis were even founded. Since the authors use discharge diagnoses to identify patients, can the authors describe how they justified the use of these cohorts in this context and how they identified sepsis diagnoses before the original consensus definition that was formulated in 1991?

REPONSE: This is an excellent point. The reviewer is correct that secular trends have influenced the coding of infection and organ dysfunction. However, given the goals of the study, the structure of the data sets, and the availability of baseline subject information, we believe that the ARIC and CHS cohorts represent the best available data sets for validating the REGARDS severe sepsis risk score. We would certainly incorporate a more recent cohort if we could identify an appropriate alternative. We amplified this point in the Limitations para 3 of the paper; “We acknowledge that the ARIC and CHS data used for validation are 10-20 years old; validation with more recent data is an important future direction.”

Reviewer 3 Report

Thank you for giving me the opportunity to review the manuscript titled "Validation of the REGARDS Severe Sepsis Risk Score". This is a good addition of previous work done on REGARDS. I have following comments for revision:
1) What is the need to have a community based validation of  the REGARDS Severe Sepsis Risk Score?
2) Why a 10 year prediction score is needed? Why not 1 year?
3) What will be the public health use of  REGARDS Severe Sepsis Risk Score validation?
4) It would be better to report PPV, NPV along with the C-statistic for the score for the benefit of the readers.
5) Another suggestion is to  look at the partial area under the ROC curve (pAUC) (i.e., true negative rate) as it is clinically more relevant than examining the entire C-statistic

Author Response

REVIEWER 3

Thank you for giving me the opportunity to review the manuscript titled "Validation of the REGARDS Severe Sepsis Risk Score". This is a good addition of previous work done on REGARDS. I have following comments for revision:

1)    What is the need to have a community based validation of the REGARDS Severe Sepsis Risk Score?
RESPONSE: We conceptualize that prevention could play a role in reducing the societal burden of severe sepsis, much as has been done for myocardial infarction and stroke. The prediction of severe sepsis risk is an important step towards establishing the framework for community prevention. We have derived a risk prediction score using the REGARDS cohort. It is now important to validate the score in independent cohorts.
    We outlined this rational in the introduction. The reviewer’s comment did cause us to pause and reflect upon whether we made a good case for the current paper. We did consider rewriting the introduction but arrived at the conclusion that the current text best makes the case for the study. If acceptable to the journal, we propose leaving the introduction as is, without further modification.
2)    Why a 10 year prediction score is needed? Why not 1 year?
RESPONSE: The reviewer makes an interesting suggestion, that perhaps 1-year risk may make a more sensible outcome measure. However, the context of our SSRS derivation is to predict long-term sepsis risk; in this context, predicting sepsis events 10 years in the future makes sense.  If we were trying to identify POST-sepsis risk (for example, risk of myocardial infarction after sepsis hospitalization), then a 1-year risk frame would make sense for a prediction rule. To avoid confusion, we opted not to add additional comment to the Limitations.
3)    What will be the public health use of  REGARDS Severe Sepsis Risk Score validation?
RESPONSE: Prediction of health risk is an important public health function. Reduction of societal cardiovascular risk has been successful because of the ability to identify the highest risk individuals, offering the opportunity to target prevention efforts at the most vulnerable individuals. A similar approach could be helpful for mitigating the societal burden of sepsis.
    We believe that Discussion paragraph 3 articulates these points. To better clarify, we added to Discussion para 3; “The SSRS may also have important public health application, identifying community groups or clusters for organized deployment of sepsis prevention measures such as those described above.”
            4) It would be better to report PPV, NPV along with the C-statistic for the score for the benefit of the readers.
RESPONSE: We thank the reviewer for this helpful comment. We have now reported the percent positive within each risk group. This helped to demonstrate the differences in the proportions of participants developing severe sepsis over ten years across the very low and very high risk groups. (See new Table 5)
            5) Another suggestion is to look at the partial area under the ROC curve (pAUC) (i.e., true negative rate) as it is clinically more relevant than examining the entire C-statistic
RESPONSE: We have added estimates of the true positive and true negative rates across the risk groups to assess where participants with and without sepsis over ten years of follow-up were classified at baseline. (See new Table 5)

Round  2

Reviewer 1 Report

Dear author

This manuscript is meaningful in terms of being described the long-term prevalence of severe sepsis. As authors’ comments, re-derivation of the SSRS with a broader cohort is necessary to generalize.

Additionally, the symbols of figure 2 ( very low group) is not visible. It need to modify

Author Response

    This manuscript is meaningful in terms of being described the long-term prevalence of severe sepsis. As authors’ comments, re-derivation of the SSRS with a broader cohort is necessary to generalize.
RESPONSE: As noted previously, re-derivation of the SSRS is beyond the scope of this effort. The current paper demonstrates that the original score has utility when using hospital discharge diagnoses. This is in fact the novelty of the current work - This score is versatile and can be applied to a range of data sets. We have commented extensively on this point on page 12 of the Limitations section of the existing paper.
    Additionally, the symbols of figure 2 ( very low group) is not visible. It need to modify
RESPONSE: We are providing a high resolution version of the figure, which will enhance its readability when the paper is published.

Reviewer 2 Report

All reviewer comments were addressed. No further suggestions.

Author Response

Thank you.